

# Substantial changes in Gaseous pollutants and health effects during COVID-19 in Delhi, India

Bhupendra Singh[1,2], Puneeta Pandey[3], Saikh Mohammad Wabaidur[4], Ram Avtar[5], Pramod Kumar[6] and Shakilur Rahman[7]

[1] Delhi School of Climate Change and Sustainability (Institute of Eminence), University of Delhi, New Delhi, Delhi, India
[2] Deshbandhu College, Department of Environmental Science, University of Delhi, New Delhi, Delhi, India
[3] Department of Environmental Sciences and Technology, Central University of Punjab Bathinda, Bathinda, Punjab, India
[4] Chemistry Department, King Saud University, Riyadh, Riyadh, Saudi Arabia
[5] Faculty of Environmental Earth Science, Hokkaido University, Sapporo, Japan, Sapporo, Japan
[6] Department of Chemistry, Sri Aurobindo College, University of Delhi, New Delhi, Delhi, India
[7] Department of Medical Elementology and Toxicology, Jamia Hamdard, New Delhi, India

Corresponding author
Bhupendra Singh,
bpsingh0783@gmail.com

## ABSTRACT

**Background**. Coronavirus disease has affected the entire population worldwide in terms of physical and environmental consequences. Therefore, the current study demonstrates the changes in the concentration of gaseous pollutants and their health effects during the COVID-19 pandemic in Delhi, the national capital city of India.

**Methodology**. In the present study, secondary data on gaseous pollutants such as nitrogen dioxide ($NO_2$), carbon monoxide (CO), sulfur dioxide ($SO_2$), ammonia ($NH_3$), and ozone ($O_3$) were collected from the Central Pollution Control Board (CPCB) on a daily basis. Data were collected from January 1, 2020, to September 30, 2020, to determine the relative changes (%) in gaseous pollutants for pre-lockdown, lockdown, and unlockdown stages of COVID-19.

**Results**. The current findings for gaseous pollutants reveal that concentration declined in the range of 51%–83% (NO), 40%–69% (NOx), 31%–60% ($NO_2$), and 25%–40% ($NH_3$) during the lockdown compared to pre-lockdown period, respectively. The drastic decrease in gaseous pollutants was observed due to restricted measures during lockdown periods. The level of ozone was observed to be higher during the lockdown periods as compared to the pre-lockdown period. These gaseous pollutants are linked between the health risk assessment and hazard identification for non-carcinogenic. However, in infants (0–1 yr), Health Quotient (HQ) for daily and annual groups was found to be higher than the rest of the exposed group (toddlers, children, and adults) in all the periods.

**Conclusion**. The air quality values for pre-lockdown were calculated to be "poor category to "very poor" category in all zones of Delhi, whereas, during the lockdown period, the air quality levels for all zones were calculated as "satisfactory," except for Northeast Delhi, which displayed the "moderate" category. The computed HQ for daily chronic exposure for each pollutant across the child and adult groups was more than 1 (HQ > 1), which indicated a high probability to induce adverse health outcomes.

## INTRODUCTION

The Coronavirus disease 2019 (COVID-19) pandemic has affected the entire population worldwide in terms of physical and mental health, the economy, and the environment. Although the effects of COVID-19 are global, the impact has been severe among low-income group strata of society. This has particularly been the case in India, where mass migration of domestic and industrial laborers and agricultural field workers from metropolitan cities to their native villages was observed; despite the government's restrictions during the countrywide lockdown. This has also resulted in the spread of COVID-19 to the country's most remote corners.

The COVID-19 pandemic has led to severe impacts on the health conditions of people across the globe. Reports from the World Health Organization and other agencies, as of September 25, 2022, indicate that COVID-19 has infected more than 620 million people, and more than 6.5 million people have died across countries (including India) (*Worldometers, 2022*). India was the second most affected country globally, following the United States of America (USA), as of September 25, 2022, with more than 44 million infected cases and more than 0.52 million deaths (*Worldometers, 2022*). Although many patients are asymptomatic, some symptoms include severe interstitial pneumonia with acute respiratory distress syndrome (ARDS) (*Ciceri et al., 2020*) and atopic dermatitis (*Patruno et al., 2020*). Further, the authors also reported a decrease in mortality in patients admitted in hospitals with COVID-19 pneumonia. The transmission is generally person-to-person, and the most likely pathway seems to be airborne (*Faridi et al., 2020*).

Various studies have reported a decrease in pollutant levels across the world during lockdown periods (*Chu et al., 2021*; *Kumari & Toshniwal, 2020*; *Sarkis et al., 2020*; *Zowalaty, Young & Järhult, 2020*; *Sahoo et al., 2020*). *Chu et al. (2021)* reported a decrease of $PM_{10}$ by 53%, 50%, and 30% in Wuhan city, Hubei Province (Wuhan excluded), and China (Hubei excluded), respectively, while $SO_2$ and $CO$ concentrations were found to be marginally reduced. *Sahoo et al. (2020)* have also reported the reduction of $PM_{2.5}$ and $PM_{10}$ levels (up to $-52$ and $-53.5\%$, respectively) during lockdown 1.0 as compared to pre-lockdown; but their level rose during the last phase of lockdown and unlock-down phases, possibly due to increased traffic return on the roads. In addition, *Kumari & Toshniwal (2020)* reported the concentration of $PM_{10}$, $PM_{2.5}$, $NO_2$ and $SO_2$ reduced by 55%, 49%, 60% and 19%, and 44%, 37%, 78% and 39% for Delhi and Mumbai, respectively, during post-lockdown phase. Several recent studies and their findings on air pollutants have been reported worldwide during the lockdown period (*Sarkis et al., 2020*; *Zowalaty, Young & Järhult, 2020*) (Table S1).

However, in the later stages of the lockdown, traffic increased when relaxation was provided for various activities, increasing the air pollution levels. Air pollution is associated with higher mortality of COVID-19 patients, with varying effects on severe acute respiratory

and cardiovascular diseases (*Domingo & Rovira, 2020*). Further, older people are among the most sensitive groups (*Lee, Son & Cho, 2007*; *Kotaki et al., 2019*). It is believed that PM forms condensation nuclei for viral attachment (*Lee et al., 2014*). A significant positive correlation was found between COVID-19 incidence and AQI ($PM_{2.5}$ and $NO_2$) in Wuhan and Xiao Gan. *Abdullah et al. (2020)* reported that $PM_{2.5}$ concentrations dominated the Air Pollutant Index (API) in Malaysia, and reductions in $PM_{2.5}$ concentrations during Malaysia Movement Control Order (MCO) were observed.

To prevent the spread of novel coronavirus (SARS-CoV2), a large number of human populations were confined to their homes during the lockdown period. This has been known as the 'Global Human Confinement Experiment', where humans' positive and adverse impacts on natural ecosystems have been evaluated (*Bate et al., 2020*). They were of the opinion that the confinement experiment could lead to a reduction in $CO_2$ emissions, pollution, and conservation of biological diversity. However, there are bound to be certain limitations and gaps in data series due to a lack of field monitoring.

Several studies reported in Indian cities that a significant relationship between air pollutants and health risk assessment in urban areas (*Kumar et al., 2014*; *Kumar et al., 2014a*; *Singh et al., 2014*; *Singh et al., 2016*; *Singh, Kumar & Jain, 2021a*; *Singh, Kumar & Jain, 2021b*). A study conducted in Delhi reported that anthropogenic VOC contribution varied between 60–70% in traffic intersection sites (*Kashyap et al., 2019*). A similar study claimed that workplace cancer risks were reported highest at traffic intersections, followed by industrial sites (*Kumar et al., 2020*; *Singh et al., 2022b*). Although various studies have been reported for early phases of lockdown in countries such as China, Italy, USA (*Dutheil, Baker & Navel, 2020*; *Tobías et al., 2020*; *Xu et al., 2020*); and a few studies on metropolitan cities of India such as Delhi and Mumbai (*Jain & Sharma, 2020*; *Mahato, Pal & Ghosh, 2020*; *Sharma et al., 2020*; *Sahoo et al., 2020*; *Singh et al., 2021c*); however, a detailed analysis involving various phases of lockdown and unlockdown is essential. Apart from this, understating the relationship between the gaseous pollutant's concentration and the non-cancer risks of short exposure to these pollutants in different age groups has become a significant research area. Hence, the objective of the present study is to evaluate the levels of gaseous pollutants in different stages of lockdown and unlockdown, thus, studying the air quality status of Delhi during these phases and estimating the air quality index (AQI). Further, during these phases, the health risk exposure (chronic, acute, and hazard quotient) is estimated for different age groups in Delhi.

## MATERIALS AND METHODS

### Data and sources

In the present study, secondary data on gaseous pollutants such as nitrogen dioxide ($NO_2$), Nitrogen oxide (NO), oxide of Nitrogen (NOx), sulfur dioxide ($SO_2$), ammonia ($NH_3$), and ozone ($O_3$) were collected from the Central Pollution Control Board (CPCB) on a daily basis. To determine the relative changes (%) in gaseous pollutants for pre-, lockdown, and unlockdown stages, data from January 1, 2020, to September 30, 2020, was collected from CPCB (https://app.cpcbccr.com/ccr#/caaqm-dashboard-all/caaqm-landing) as previously described in *Singh et al. (2022)*; *Singh et al. (2022a)*; *Singh et al. (2022b)*.

In the present study, the hourly and daily data pertaining to the concentration of gaseous pollutants were obtained from the Central Pollution Control Board (CPCB). While procuring continuous data on these gaseous pollutants, a few of the hourly and daily data were found missing. As a result, mean substitution was used, where average values were used to replace those missing values. Further, the daily average of measurements was calculated at every station. The technical specifications of the measurements and instruments used can be found on the CPCB website (https://cpcb.nic.in/archivereport.php). The CPCB database establishes a meticulous approach for sampling, analysis, and calibration; thus, furnishing data quality assurance (QA) or quality control (QC) programs (*Gurjar, Ravindra & Nagpure, 2016*; *Singh, Kumar & Jain, 2021b*).

AQI India provided air pollution data with real-time AQI for various air pollutants. The National Air quality standard (NAAQS) revised its criteria according to eight parameters, namely, $PM_{10}$, $PM_{2.5}$, $NO_2$, $SO_2$, CO, $O_3$, $NH_3$, and Pb, for the short-term (up to 24 h on average) periods (CPCB, 2016). The AQI was used to retrieve information on air quality in terms of pollution levels because it is directly associated with public health safety. A total of six AQI categories, namely, Good, Satisfactory, Moderate, Poor, Very Poor, and Severe, have been defined as associated with health risks.

## Measures

The pandemic situation was classified into three periods: before lockdown, during the lockdown, and after unlockdown. The time before lockdown (between January 1, 2020, and March 24, 2020) was termed the 'pre-lockdown' period, whereas the time between March 25 and May 31, 2020, was termed the 'during lockdown period.' As a result, lockdown was extended four times until May 30, 2020, with the first, second, third, and fourth phases ending on April 14, 2020 (lockdown I), 3rd May 2020 (lockdown II), 17th May 2020 (lockdown III), May 30 (lockdown IV), respectively. After this, the government started relaxing the lockdown and started opening up avenues in a restricted manner during various phases of unlockdown starting from 1st to June 30, 2020 (unlockdown I), for the period of 1st to July 31, 2020 (unlockdown II), 1st to August 31, 2020 (unlockdown III), and 1st to September 30, 2020 (unlockdown IV). The continuous reduction in the levels of gaseous pollutants except ozone was observed in subsequent months during the complete lockdown caused by the complete closure of markets and industrial activities.

All 21 air quality monitoring stations in Delhi were selected in this study and classified into different districts (https://ceodelhi.gov.in/OnlineErms/Reports/PS_LocationListOn15thOctDistrictwise.aspx; Table S2). These include Ashok Vihar, Jawaharlal Nehru Stadium (JLN Stadium), Arya Nagar, Karni Singh, Delhi Technical University (DTU), Income Tax Office (ITO), Jahangirpuri, Lodhi Road, Najafgarh, Narela, North Campus University, Ramakrishna Puram (RK Puram), Rohini, Vivek Vihar, Siri Fort, Okhla, Patparganj, Bawana, Sri Aurobindo Marg, Alipur and Wazirpur.

## Human health risk assessment

Human Health Risk Assessment (HHRA) is a process by which the potential impact of the toxin on humans due to exposure is characterized. *Morakinyo, Mukhola & Mokgobu*

*(2020)* assessed the possible health risks from exposure to gases such as $SO_2$, $NO_2$, CO, and $O_3$ using the US Environmental Protection Agency's (EPA) human health risk assessment framework. The HHRA is a tool used by regulatory agencies to assist in the formulation of policies that protect public health against the harmful effects of air pollution (*Organisation for Economic Cooperation and Development , 2008*).

### Hazard identification

Hazard identification is a procedure to identify the pollutant in an ambient environment that is likely to induce harmful effects on human health (*Saliba et al., 2016*). The identification of gaseous pollutants such as $SO_2$, $NO_2$, CO, and $O_3$ that pose a risk to human health was performed through a review of existing literature (*United States Environmental Protection Agency, 2009*; *Gilbert et al., 2019*; *Kim, Kang & Kim, 2018*; *Odekanle et al., 2020*).

### Exposure assessment

The exposure assessment calculates the population exposed to the pollutant, the magnitude, and the duration of exposure. In the present study, the inhalation route is the major route of exposure to the identified pollutants. We estimated the daily and annual readings for normal and acute exposure periods for different age groups, namely infants (birth to a year), toddlers (2–5 years), children (6–12 years), and adults (19–75 years). The classification of age groups has been carried out according to Morakinyo et al. (2017, 2020).

For exposure to non-carcinogenic pollutants ($SO_2$, $NO_2$, and $O_3$), the acute exposure rate equation is mentioned as:

$$AHD = C \times IR/BW \tag{1}$$

where AHD is the average hourly dose for inhalation ($\mu g/kg/hour$), C is the concentration of the pollutant ($\mu g/m^3$), IR is the inhalation rate ($m^3/h$), and BW is the body weight (kg) (*World Health Organization, 1999*). For exposure to non-carcinogenic pollutants ($NO_2$, $SO_2$, $O_3$), the chronic exposure equation used for the inhalation exposure route is:

$$ADD = (C \times IR \times ED)/(BW \times AT) \tag{2}$$

where, ADD is the average daily dose of the pollutant ($\mu g/kg/day$), ED is the exposure duration (days), and AT is the averaging time (days) (USEPA, 1997).

$$ED \text{ (exposure duration)} = ET \times EF \times DE \tag{3}$$

where, ET is the exposure time (hour/day), EF is the exposure frequency (days/year), and DE is the duration of exposure (year). The recommended values in equations of the daily exposure dose of $NO_2$, $SO_2$, and $O_3$ are presented in Table 1.

### Risk characterization

Risk characterization is an estimation of risk to human health due to exposure to a pollutant. In this study, the health risk estimate of possible non-carcinogenic exposure to the pollutant was determined using the hazard quotient (HQ) method (*Muller et al., 2003*; *Saliba et al., 2016*).

**Table 1 The recommended values in equations of the daily exposure dose of $NO_2$, $SO_2$, $O_3$.**

| Parameter | Definition | Values for age categories | | | | Reference |
| | | Infant (0–1 yr.) | Toddler (2–5 yrs.) | Child (6–12 yrs.) | Adult (19–75 yrs.) | |
|---|---|---|---|---|---|---|
| EF | Exposure frequency (days/year) | 350 | 350 | 350 | 350 | *Morakinyo et al. (2017)* |
| ED | Exposure frequency (days/year) | 1 | 6 | 12 | 30 | *USEPA (1992)* |
| AT | Averaging time (days); AT = ED X365 days | 365 | 2190 | 4380 | 10950 | *Matooane & Diab (2003)* |
| BW | Body weight (kg) | 11.3 | 22.6 | 45.3 | 71.8 | *Morakinyo et al. (2017)* |
| InhR | Inhalation rate (m3/day) | 9.2 | 16.74 | 21.02 | 21.4 | *Morakinyo et al. (2017)* |

The risk to health due to exposure to CO, $NO_2$, $SO_2$, and $O_3$ through the inhalation route was estimated using Eqs. (4) and (5)

$$ADD\ inh = \frac{C \times InhR \times EF \times ED}{BW \times AT} \tag{4}$$

$$HQ = \frac{ADD}{REF}$$

or

$$HQ = x = \frac{AHD}{REF} (acute\ exposure) \tag{5}$$

REF is the reference exposure level for an exposed pollutant that has significant adverse health effects that are likely to occur in exposed groups compared to the unexposed group. Therefore, the RELs for the different pollutants are prescribed by NAAQS. For NO2 and SO2, these are 188 $\mu$g/m$^3$ and 125 $\mu$g/m$^3$ for daily and 40 $\mu$g/m$^3$,50 $\mu$g/m$^3$ for annual exposure besides 120 $\mu$g/m$^3$ $O_3$ of 8 h duration (*Morakinyo, Mukhola & Mokgobu, 2020*).

## Statistical analysis and procedure

The present study evaluated the variability in air quality parameter concentrations and associated health assessment for 21 monitoring stations in Delhi for pre-lockdown, during the lockdown, and unlockdown periods. The time series plotting technique for particulate and gaseous pollutants was used to determine variable changes during the pre-lockdown, during lockdown, and unlockdown periods. In the present study, SPSS version 26.0 with the significant level "$p < .05$" was employed to carry out the statistical analyses. Further, the correlation was computed to Pearson correlation analysis was performed to find out closely associated variables. ArcGIS 10.4.1 was used to identify the changes in variables and generate an AQI map during the lockdown period as compared to the pre-lockdown and post-lockdown periods.

## RESULTS AND DISCUSSION

### Concentration of gaseous pollutants in the Pre-lockdown, during the lockdown, and unlockdown periods

In the present study, emphasis was laid on determining changes in the concentrations of gaseous air pollutants comprising NO, $NO_2$, NOx, $O_3$, $NH_3$, and $SO_2$, in Delhi during the

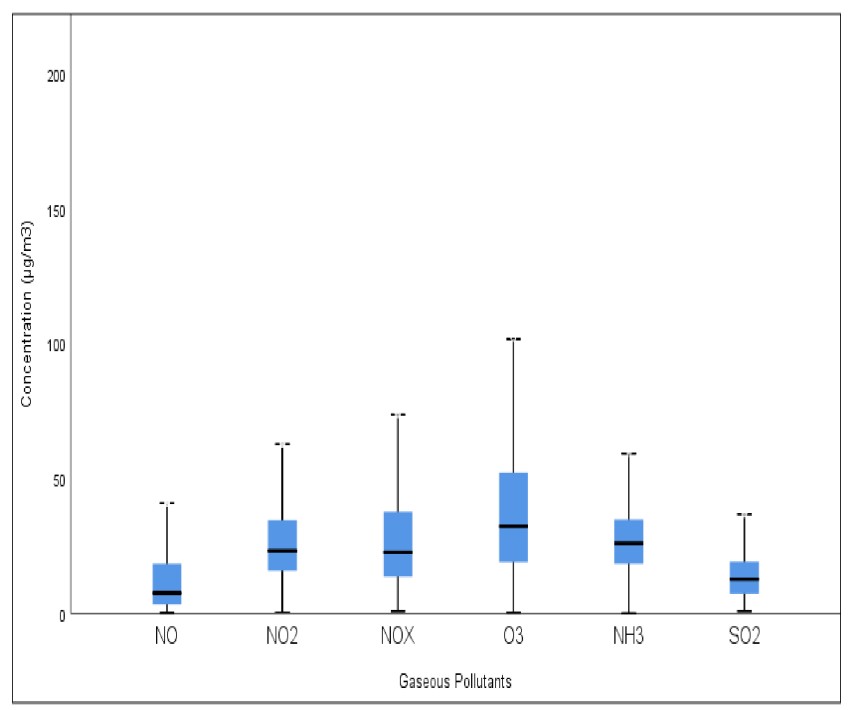

**Figure 1 Box plot for all pollutants at Delhi during a pandemic situation.**

pandemic. It was observed that there was a decrease in the concentration of all gaseous pollutants except $O_3$ from January 2020 to September 2020. Further, in subsequent months, a continuous decline in the concentration of these gaseous pollutants was observed as a result of the complete lockdown. It is believed that as more and more people were forced to spend time in their homes, the COVID-19 pandemic changed the dynamics of anthropogenic emissions from various activities (*Gautam et al., 2020*). The average concentrations across all stations were found to be 16.99, 27.68, 31.25, 38.55, 27.98, and 17.10 µg/m$^3$ for NO, $NO_2$, $NO_x$, $O_3$, $NH_3$, and $SO_2$, during the pandemic period, respectively (Fig. 1). The average values for gaseous pollutants was found to be 38.09, 21.07, 22.41, 24.33, 27.07, 24.15, 23.46, 24.22, and 29.49 µg/m$^3$ for $NO_2$, 48.77, 15.53, 23.52, 23.02, 23.97, 20.88, 19.32, 23.73, and 32.70 µg/m$^3$ for $NO_x$, 31.93, 48.64, 55.55, 65.41, 54.50, 41.29, 34.53, 26.49, and 33.54 µg/m$^3$ for $O_3$, 32.66, 23.42, 30.22, 26.48, 24.56, 27.82, 26.85, 23.53, and 24.71 µg/m$^3$ whereas for $NH_3$ 15.99, 21.12, 21.61, 2.87, 24.93, 17.93, 17.93, 14.27, 13.09, and 14.33 µg/m$^3$ during the pre-lockdown, lockdown I, lockdown II, lockdown IV, unlockdown I, unlockdown II, unlockdown III, and unlockdown IV, respectively.

During the pandemic situation, the time-series graph for NO, NO2, NOx, O3, NH3, and SO2 has been presented in Fig. 2 for pre-lockdown, lockdown I, lockdown II, lockdown III, lockdown IV, unlockdown I, unlockdown II, unlockdown III, and unlockdown IV. During the lockdown period, the decline in nitrogen oxide ($NO_x = NO + NO_2$) witnessed a sharp reduction in emissions from both public and private transport (*Kroll et al., 2020*). The values of $NO_2$, $NO_x$, and $NH_3$ were reduced by 38%, 56%, and 20%, whereas the value

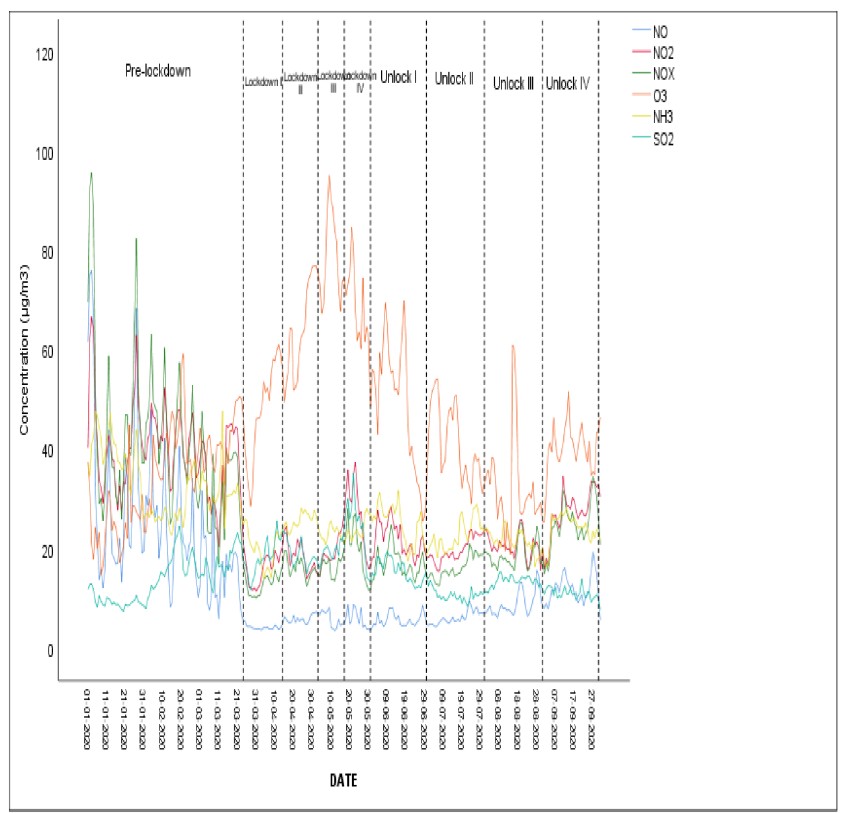

**Figure 2** Time series graph for all pollutants at Delhi during a pandemic situation.

of $O_3$ was increased by 75% during the lockdown period as compared to the pre-lockdown period. 20% reported similar results in China for 30–50 days (*Li et al., 2020a*; *Li et al., 2020b*; *Zinke, 2020*). The maximum values for NO, NO2, NOx, O3, $NH_3$, and $SO_2$ were found at Lodhi Road, North Campus University, Siri Fort, and Najafgarh sites during the pandemic situation. The ozone level during the lockdown period was observed to increase in concentration. Nitrogen oxides (NO, $NO_2$) and VOCs in the presence of sunlight in the atmosphere are responsible for ozone formation (*Zheng et al., 2021*). The ozone concentration was observed to be higher as compared to other pollutants during the lockdown period because of the absence of NO formation from transport emissions which could not dissociate into $O_2$ and O. The high level of ozone is usual during the spring and summer sessions due to higher solar radiation, which promotes the photocatalysis of $NO_2$ (*Collivignarelli et al., 2020*). A slight increase in the concentration of $SO_2$ was observed during the lockdown period compared to the unlock-down period. This could be due to the restricted supply of power plants in northern India compared to using coal-powered plant energy—an essential commodity during the lockdown.

## Zone-wise level of gaseous pollutants

### New Delhi (ND)

The monitoring station of RK Puram is represented as the New Delhi zone. The average concentration of RK Puram station was 18.91, 34.79, 34.03, 31.09, and 22.61 $\mu g/m^3$ for NO, $NO_2$, NOx, $O_3$, and $NH_3$, respectively (Fig. 3A). The average concentrations for NO, $NO_2$, NOx, $O_3$, and $NH_3$ were calculated to be 42.62, 46.06, 59.08, 24.59, and 17.10 $\mu g/m^3$ for pre-lockdown whereas they were 3.58, 18.71, 12.85, 55.53 and 19.99 $\mu g/m^3$ for lockdown I, 3.56, 22.15, 14.64, and 28.16 $\mu g/m^3$ for lockdown II, 2.11, 20.92, 12.85, 58.05 and 31.28 $\mu g/m^3$ for lockdown III, whereas 4.78, 38.52, 24.39, 66.39, and 19.24 $\mu g/m^3$ for lockdown IV, respectively. The average concentrations for unlockdown I, unlockdown II, unlockdown III, and unlockdown IV were observed to be 4.42, 7.36, 10.86, 21.06 $\mu g/m^3$ for NO, 28.70, 32.89, 30.40, 38.99 $\mu g/m^3$ for NO2, 18.94, 23.43, 30.40, 38.99 $\mu g/m^3$ for NOx, whereas, 24.75, 35.18, 20.24, 21.60 $\mu g/m^3$ for NH3, respectively. The average values of ozone were found to be 19.35, 10.79, 28.25, and 31.38 $\mu g/m^3$, respectively. The level of gaseous pollutants witnessed a sharp decline during the first two of the four phases of lockdown corresponding to the last year, except for the ozone level. Further, the role of meteorological parameters, especially rainfall, has been considered a significant factor in reducing the concentration and washing out of various atmospheric pollutants (*Elperin et al., 2011*; *Guo et al., 2016*; *Bedi et al., 2020*)

### South Delhi (SD)

Karni Singh Marg, Jawaharlal Nehru Stadium, Lodhi Road, and Sri Aurobindo Marg were four monitoring stations under the South Delhi zone. The average concentrations were found to be 20.59 $\mu g/m^3$ (NO), 27.47 $\mu g/m^3$ (NO2), 34.49 $\mu g/m^3$ (NOx), 46.79 $\mu g/m^3$ ($O_3$), 27.45 $\mu g/m^3$ ($NH3$), and 12.22 $\mu g/m^3$ ($SO_2$), respectively (Fig. 3B). The average concentrations for NO, $NO_2$, NOx, $O_3$, $NH_3$, and$SO_2$ were calculated to be 10.48, 28.27, 23.57, 62.46, 23.57, and 11.29 $\mu g/m^3$ for Karni Singh Marg, respectively. The average concentrations for NO, $NO_2$, $NO_x$, $O_3$, $NH_3$, and$SO_2$ were found to be 24.70, 37.74, 41.04, 28.33, and 18.01 $\mu g/m^3$ for Jawaharlal Nehru Stadium, respectively, whereas for Lodhi road, it was 41.25, 18.54, 56.93, and 30.53 $\mu g/m^3$ for NO, $NO_2$, NOx, and $O_3$ respectively. The average trend for all South Delhi zone monitoring stations was observed in Lodhi road > JLN Stadium > Karni Singh Marg > Sri Aurobindo Marg for NO, $NO_2$, and $NO_x$ during the pandemic situation, respectively. In the case of ozone, the trend was shown as Karni Singh Marg > JLN Stadium > Lodhi road > Sri Aurobindo Marg during the lockdown period, respectively. The average ozone level trend was observed to increase with subsequent lockdowns: lockdown I > lockdown II > lockdown III > lockdown IV, respectively. In contrast with the absence of dissociation factors (NO and NO2) in the ambient atmosphere, ozone levels at all stations showed a slightly increasing trend during the lockdown period, similar to those during the pre-lockdown period (*Dhaka et al., 2020*). The ozone level was higher at Karni Singh Marg than at other stations in South Delhi, possibly due to more restrictions on transport activities on these premises.

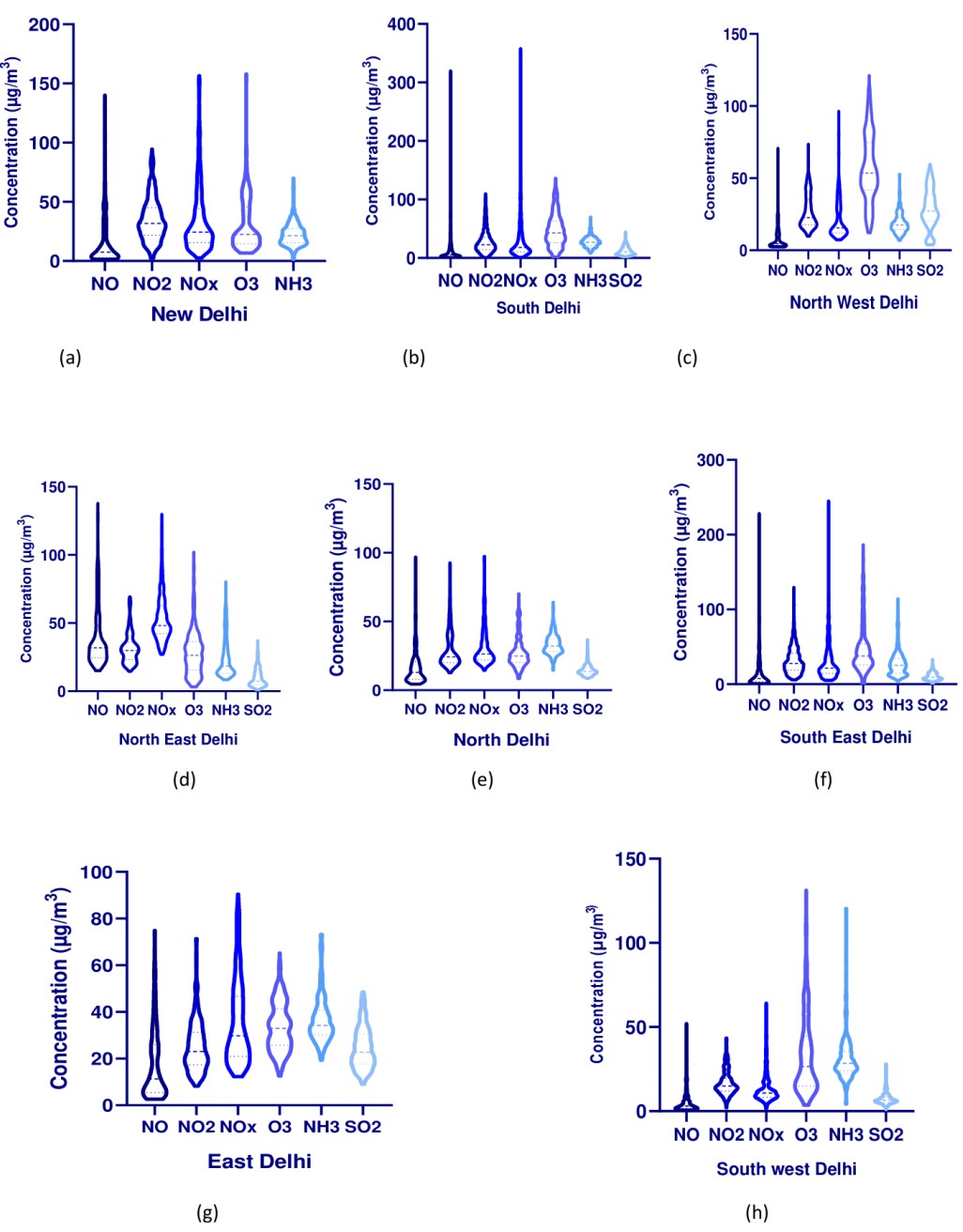

**Figure 3** (A–H) Violin plots for all pollutants zone-wise in Delhi during a pandemic situation.

### Northwest Delhi (NWD)

The monitoring stations such as Rohini, Ashok Vihar, and DTU were considered under the North-west Delhi. The average values for Rohini, Ashok Vihar, and DTU were 9.11 µg/m³ (NO), 27.06 µg/m³ (NO₂), 21.60 µg/m³ (NOx), 56.86 µg/m³ (O₃), 18.98 µg/m³ (NH₃), and 28.69 µg/m³ (SO₂) respectively, a similar result was reported in Delhi (*Dutta & Jinsart, 2020*) (Fig. 3C). The average trend of gaseous pollutants was found to be Rohini

> DTU > Ashok Vihar, respectively. The average trend for gaseous pollutants at Rohini declined during the lockdown period and again increased continuously in the unlock-down period, except for $O_3$. The average concentration of $O_3$ was continuously increased with values 38.57, 80.75, 98.19, 110.88 $\mu g/m^3$ during the pre-lockdown period, lockdown I, lockdown II, lockdown III, and lockdown IV, respectively. The trend of mean values of gaseous pollutants for Rohini was found to be $O_3$ > $NO_2$ > $NH_3$ > $NO_x$ > $SO_2$ > NO. A similar trend was observed for DTU and Ashok Vihar, respectively. North-west Delhi zone is considered as the industrial zone where the decrease in $NO_2$ concentration could be attributed to the substantial reduction in emissions, mainly from transport sectors and industrial processes.

### Northeast Delhi (NWD)

The monitoring station of ITO has been represented under the Northeast Delhi zone. The average concentrations of gaseous pollutants were calculated to be 39.53, 31.38, 53.52, 28.12, 24.75, and 9.78 $\mu g/m^3$ for NO, $NO_2$, NOx, $O_3$, $NH_3$, and $SO_2$, respectively (Fig. 3D). The average trend of gaseous pollutants for ITO observed a sharp decline in lockdown I > lockdown II > l lockdown III > lockdown IV during the lockdown period. In contrast, the unlockdown period showed an increasing trend in the order of unlockdown I > unlockdown II > unlockdown III > unlockdown IV, respectively, except $O_3$. Ozone levels during the lockdown period witnessed an increase as a result of lowering the concentration of NOx, which may result in ozone intensification levels in urban areas (*Mahato, Pal & Ghosh, 2020*). In addition, a natural increase in insolation and temperature from March to August in the northern hemisphere might lead to a rise in the value of ozone when the maximum ozone concentration is usually recorded (*Gorai, Tchounwou & Mitra, 2017*).

### North Delhi zone (NoD)

The monitoring stations such as Alipur, Narela, Bawana, Wazirpur, and North Campus were considered under the North Delhi zone. The average concentrations for gaseous pollutants were calculated to be 15.51, 28.43, 29.44, 27.83, 30.91, and 14.53 $\mu g/m^3$ for NO, $NO_2$, NOx, $O_3$, $NH_3$, and $SO_2$, for the North Delhi zone, respectively (Fig. 3E). The average values for NO, $NO_2$, NOx, $O_3$, and $NH_3$ for Alipur were observed to be 3.26, 23.44, 15.13, 52.65, and 20.51$\mu g/m^3$ during the lockdown period, whereas 5.05, 14.09, 11.37, 17.34, 26.66 $\mu g/m^3$ during the unlockdown period. The level of pollutants slightly increased during the unlockdown phase, except for ozone; *Dutta & Jinsart (2020)* noticed a similar result in Delhi. The average values for NO, $NO_2$, NOx, $O_3$, and $NH_3$ for Narela were observed to be 4.17, 31.13, 19.39, 70.73, 34.99, and 15.86 $\mu g/m^3$ during the lockdown period, whereas 8.35, 27.73, 21.55, 21.06, 17.06, and 5.86 $\mu g/m^3$ during the unlockdown period. The average concentrations for NO, $NO_2$, NOx, $O_3$, $NH_3$, and $SO_2$ for Wazirpur were found to be 8.21, 33.32, 24.33, 32.31, 30.01, and 15.72 $\mu g/m^3$, whereas 21.52, 32.16, 22.85, 13.43, 35.67, 19.80 $\mu g/m^3$ during the unlockdown period, respectively. The average concentrations for NO, $NO_2$, NOx, $O_3$, and $SO_2$ for Bawana were 1.04, 22.2, 12.7, 76, and 22.5 $\mu g/m^3$, whereas 2.69, 22.5, 14.2, 34.1, 7.87 $\mu g/m^3$ during the unlockdown period, respectively. The average concentrations for NO, $NO_2$, NOx, and $O_3$, for North Campus University were reported as 10.69, 17.22, 27.91, and 18.81 $\mu g/m^3$ during the lockdown period and 17.88,

16.73, 33.13, and 32.82 $\mu g/m^3$ during the unlockdown period, respectively. The results for gaseous pollutants witnessed a sharp decline during the lockdown period, whereas it slightly increased during the unlockdown period. A similar study conducted in China reported a 6.76 and 24.67 percent reduction in $SO_2$ and $NO_2$ (*Dantas et al., 2020*).

### Southeast Delhi (SED)

The monitoring stations of Okhla and Siri Fort have been represented under the South Delhi zone. The average concentrations for Vivek Vihar were observed as 11.02, 27.08, 33.93, 36.87, 28.09, 24.60 $\mu g/m^3$ whereas, for Siri Fort, it was 13.34, 31.16, 28.32, 65.30, 19.56, 8.23 $\mu g/m^3$ for NO, $NO_2$, NOx, $O_3$, $NH_3$, and $SO_2$ during the pandemic situation respectively (Fig. 3F). The decline in NO, $NO_2$, and NOx could be mainly attributed to restrictions imposed on transport (public and private vehicular movement) during the lockdown. The mean trend for NO, $NO_2$, NOx, $O_3$, $NH_3$, and $SO_2$ was found to be lockdown III > lockdown IV > lockdown I > lockdown II>during the lockdown period, whereas for the unlock-down period, it showed an increasing trend in the order of unlockdown IV > unlockdown II > unlockdown III > unlockdown I respectively except for $O_3$.

### East Delhi (ED)

The monitoring stations of Arya Nagar, Vivek Vihar, Patparganj, and Anand Vihar have been represented in East Delhi. The average concentrations of East Delhi were found to be 13.24, 22.43, 30.19, 37.99, 37.99, 36.21, and 26.21 $\mu g/m^3$ for NO, $NO_2$, NOx, $O_3$, $NH_3$, and$SO_2$ during the pandemic period, respectively (Fig. 3G). The mean trend for NO, $NO_2$, NOx, $O_3$, $NH_3$, and $SO_2$ was found to be lockdown IV > lockdown III > lockdown II > lockdown I during the lockdown period, whereas, for the unlockdown period, it observed an increasing trend in the order of unlockdown IV > unlockdown II > unlockdown III > unlockdown I respectively. Anand Vihar has been considered one of the inter bus terminals of NCR, which found massive traffic congestion and slow movement. During the unlock-down period, the state government has announced relief measures for migrants, and arrangements for buses for migrants to return to their native place increased the level of pollution at Anand Vihar station (*Singh & Kumar, 2021*).

### Southwest Delhi (SWD)

Two monitoring stations of Najafgarh and Sri Aurobindo Marg have been represented in Southwest Delhi. The average concentrations were found to be 2.40, 9.04, 12.47, 12.08, 33.57, 19.24 $\mu g/m^3$ for NO, $NO_2$, NOx, $O_3$, $NH_3$, and $SO_2$ during the pandemic period, respectively (Fig. 3H). The mean trend for NO, $NO_2$, NOx, $O_3$, $NH_3$, and $SO_2$ was found to be lockdown I > lockdown II > lockdown II > lockdown IV during the lockdown period, whereas for the unlockdown period, it observed an increasing trend in the order of unlockdown IV > unlockdown III > unlockdown II > unlockdown I respectively. The lowest concentrations for all gaseous pollutants were observed at Sri Aurobindo Marg due to more greenery and complete restriction of anthropogenic activities such as transportation, traveling, and industrial activities, which are the primary source of such pollutants (*Sharma et al., 2020*). Other studies conducted in different cities of India indicated that, on average, $NO_x$ concentrations declined by 60%, further suggesting that the impact of lockdown

on air quality was not homogenous across the country due to fluctuating background contributions (*Kumari & Toshniwal, 2020*; *Gulia et al., 2021*).

## Non-carcinogenic health risks of $NO_2$, $SO_2$, and $O_3$ via inhalation route

The present study calculated health risk assessment and hazard identification for non-carcinogenic risks from exposure to ADD of $NO_2$, $O_3$, and $SO_2$ via the inhalation route, where we estimated that the inhalation route is the prime pathway for exposure to any pollutants. The health quotient (HQ) was calculated for each pollutant across the exposed groups (infant, toddler, child, and adult) for pre-, during, and unlock-down periods in Delhi. The calculated HQ for daily and annual exposure for each pollutant across the exposed groups was less than 1 (HQ<1) for pre, during, and unlock-down periods. This indicates that the level of pollutants ($NO_2$. $O_3$, and $SO_2$) recorded for Delhi was unlikely to pose any adverse risk to public health (infants, toddlers, children, and adults). However, in infants (0–1 year), HQ for daily and annual groups was calculated higher than the rest of the exposed group (toddler, children, and adult) in all the periods (Table 2). Thus, for all pollutants, daily and annual exposure induces more significant non-carcinogenic effects during the pre-lockdown period than during the lockdown and unlockdown period. The order of HQ for all exposed pollutants was pre-lockdown > unlockdown > lockdown for both daily and annual exposure groups except $O_3$.

Furthermore, daily and annual chronic exposure to exposed pollutants has been calculated for different exposed groups for pre-lockdown, during the lockdown, and unlockdown periods. The calculated HQ for daily chronic exposure to each pollutant across the infant group was found to be less than 1 (HQ< 1), which varies from 0.04 to 0.08 ($NO_2$), 0.09 to 0.17 ($O_3$), and 0.03 to 0.05 ($SO_2$) respectively and which indicates unlikely induced adverse health outcomes. However, the calculated HQ for daily chronic exposure for each pollutant across the child and adult groups was found to be more than 1 (HQ> 1), which varies from 2.99 to 5.54 ($NO_2$), 6.51 to 12.24 ($O_3$), and 2.51 to 3.70 ($SO_2$) for child and 2.36 to 4.37 ($NO_2$), 5.13 to 9.66 ($O_3$), and 2.51 to 2.92 ($SO_2$) for adult respectively and indicated likely to induce adverse health outcomes (Table 3). The order of HQ for exposed pollutants was pre-lockdown > unlockdown > lockdown for both daily and annual exposure groups except $O_3$; a similar observation was also noticed in Delhi (*Dutta & Jinsart, 2020*) and Kolkata (*Bera et al., 2020*). A similar trend for annual chronic exposure was observed for all exposed groups for pre-lockdown, during the lockdown, and unlockdown periods (Fig. 4).

The present study shows that the health risk occurring from exposure to the concentration of gaseous pollutants that were below the prescribed limit by the regulatory agencies (*Katsouyanni et al., 1997*). Several studies reported that human exposure to a low concentration of $NO_2$ might be responsible for respiratory infections and acute and obstructive lung diseases (*Chen et al., 2012*; *Santus et al., 2012*). Many research scholars reported a significant relationship between exposure to a slight concentration of $NO_2$ and the occurrence of acute ischaemic stroke (*Vidale et al., 2010*; *Anderson et al., 2012*). Exposure to $O_3$ could induce various health risks such as coughing, breathlessness, chest

Singh et al. (2023), *PeerJ*, DOI 10.7717/peerj.14489

**Table 2   Hazard quotients of $NO_2$, $O_3$, and $SO_2$ for daily and annual exposure groups.**

| | Pre-lockdown | | | | Lockdown | | | | Unlockdown | | | |
|---|---|---|---|---|---|---|---|---|---|---|---|---|
| | Infant (0–1 yrs) | Toddler (2–5 yrs) | Child (6–12 yrs) | Adult (19–75 yrs) | Infant (0–1 yrs) | Toddler (2–5 yrs) | Child (6–12 yrs) | Adult (19–75 yrs) | Infant (0–1 yrs) | Toddler (2–5 yrs) | Child (6–12 yrs) | Adult (19–75 yrs) |
| | | | | | | Daily exposure group | | | | | | |
| $NO_2$ | 0.16 | 0.15 | 0.09 | 0.06 | 0.09 | 0.08 | 0.05 | 0.03 | 0.10 | 0.09 | 0.05 | 0.03 |
| $O_3$ | 0.19 | 0.17 | 0.11 | 0.07 | 0.36 | 0.33 | 0.20 | 0.13 | 0.22 | 0.20 | 0.12 | 0.08 |
| $SO_2$ | 0.09 | 0.02 | 0.05 | 0.03 | 0.11 | 0.02 | 0.06 | 0.04 | 0.07 | 0.01 | 0.04 | 0.03 |
| | | | | | | Annual exposure group | | | | | | |
| $NO_2$ | 0.76 | 0.69 | 0.43 | 0.27 | 0.41 | 0.37 | 0.23 | 0.15 | 0.45 | 0.41 | 0.26 | 0.16 |
| $O_3$ | 0.19 | 0.17 | 0.11 | 0.07 | 0.36 | 0.33 | 0.20 | 0.13 | 0.22 | 0.20 | 0.12 | 0.08 |
| $SO_2$ | 0.23 | 0.04 | 0.13 | 0.08 | 0.27 | 0.05 | 0.15 | 0.10 | 0.18 | 0.03 | 0.10 | 0.07 |

Singh et al. (2023), *PeerJ*, DOI 10.7717/peerj.14489

**Table 3  Hazard quotients of $NO_2$, $O_3$, and $SO_2$ for daily and annual chronic exposure groups.**

| | Pre-Lockdown | | | Lockdown | | | Unlockdown | | |
|---|---|---|---|---|---|---|---|---|---|
| | Infant (0–1 yr) | Child (6–12 yrs.) | Adult (19–75 yrs.) | Infant (0–1 yr.) | Child (6–12 yrs.) | Adult (19–75 yrs.) | Infant (0–1 yr) | Child (6–12 yrs.) | Adult (19–75 yrs.) |
| | Daily Chronic Exposure | | | | | | | | |
| $NO_2$ | 0.08 | 5.54 | 4.37 | 0.04 | 2.99 | 2.36 | 0.05 | 3.29 | 2.60 |
| $O_3$ | 0.09 | 6.51 | 5.13 | 0.17 | 12.24 | 9.66 | 0.10 | 7.49 | 5.90 |
| $SO_2$ | 0.04 | 3.15 | 2.49 | 0.05 | 3.70 | 2.92 | 0.03 | 2.51 | 1.98 |
| | Annual Chronic Exposure | | | | | | | | |
| $NO_2$ | 0.36 | 26.05 | 20.55 | 0.20 | 14.08 | 11.10 | 0.22 | 15.47 | 12.20 |
| $O_3$ | 0.09 | 6.51 | 5.13 | 0.17 | 12.24 | 9.66 | 0.10 | 7.49 | 5.90 |
| $SO_2$ | 0.11 | 7.88 | 6.21 | 0.13 | 9.26 | 7.30 | 0.09 | 6.27 | 4.95 |

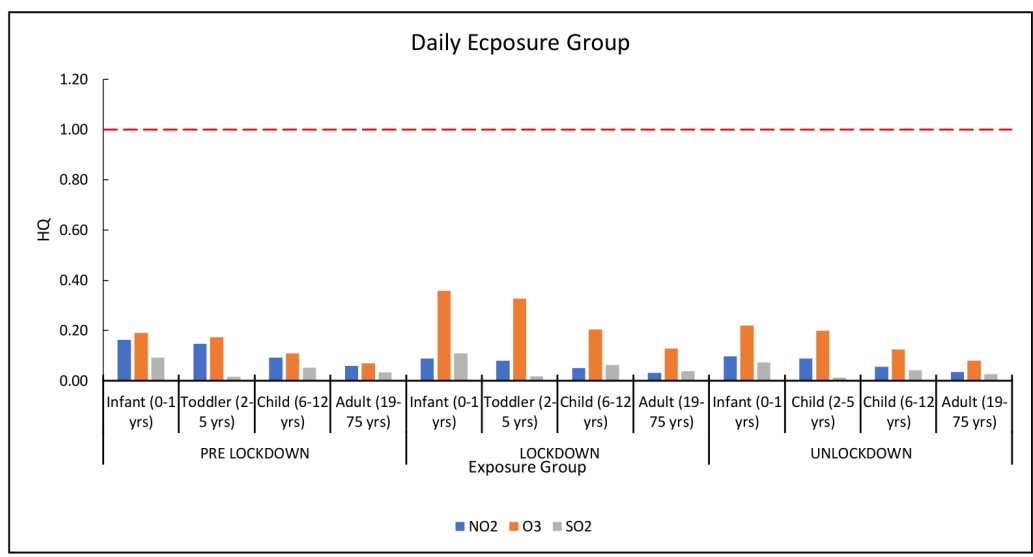

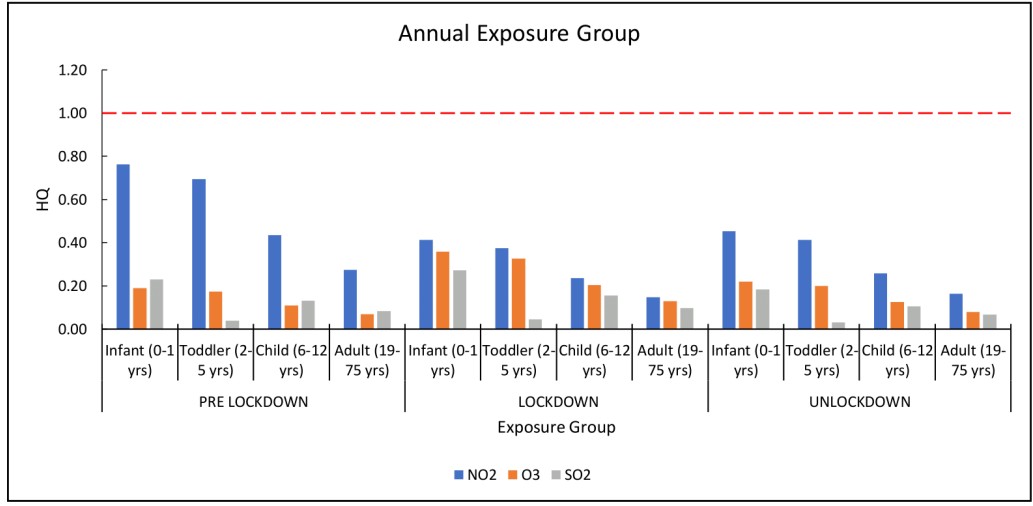

**Figure 4** Carcinogenic risks of NO$_2$, SO$_2$, and O$_3$ for daily and annual exposure group.

pain, and impaired lung tissues (*Kim, Jahan & Kabir, 2013*). The health quotient for different pollutants has been presented for all other zones of Delhi in the (Table S3).

## Air quality index (AQI)

An Air Quality Index (AQI) provides information about air pollutants, thus, indicating the air quality of a place. As a result, AQI is directly associated with the public health of society since AQI is directly proportional to the public health risk. India's ambient air quality standard is prescribed by National Ambient Air Quality Standards (NAAQS) and computed on the basis of concentration levels of eight pollutants for every 24 h. The air pollution levels have been classified into six AQI groups, namely, "Good," "Satisfactory,"

"Moderate," "Poor," "Very Poor," and "Severe. Several studies reported a significant reduction in air pollutants across 91 Indian cities during the month of March 2020, which were leading to in 'satisfactory' to 'good' category (*Dumka et al., 2020*, *Jain & Sharma, 2020*; *Navinya, Patidar & Phuleria, 2020*; *Metya et al., 2020a*; *Metya et al., 2020b*; *Manisha & Kulshrestha, 2021*). This drastic improvement in AQI was observed across the Indian sub-continent; AQI was improved in Karachi, Islamabad, and Lahore (Khurshied, 2020; *Ali et al., 2020*), Kathmandu (*Pant et al., 2020*; *Shrestha & Shrestha, 2005*), and Bangladesh (*Pavel et al., 2020*; *Islam et al., 2021*).

The air quality level had improved significantly due to reduced air pollutant levels during the lockdown period. The range of air quality levels for all zones was calculated 210 (New Delhi) and 306 (North Delhi), respectively, during the pre-lockdown period, which can be categorized as "poor-to-very-poor." The air quality values for pre-lockdown were found to be "poor category for ED, ND, SD, and SWD to the "very poor" category for NoD, NED, NWD, and SED. The air quality levels for all zones were calculated as "satisfactory" (62–96), except for NED which displayed "moderate" category (163) during the lockdown period (Fig. 5). This result indicates a drastic improvement in air quality during the lockdown period. The values for air quality levels were increased by 71% (ED), 68% (ND), 69% (NoD), 46% (NED), 70% (NWD), 75% (SD), 76% (SED), and 67% (SWD).

## Correlation

In the present study, an attempt was made to correlate the concentration of various gases with the monitoring stations of Delhi. Table S4 describes the correlation of NO with various sites in Delhi. It can be seen that NO at Alipur is strongly correlated with Bawana (0.79), DTU (0.67), JLN (0.81), Karni Singh (0.71), Patparganj (0.80), RK Puram (0.74), Rohini (0.71), while Ashok Vihar with DTU (0.62), Bawana with DTU (0.77), JLN with Karni Singh (0.81), Patparganj (0.86), RK Puram (0.81) and Rohini (0.81) reported a similar trend.

Correlation of $NO_2$ and $NO_x$ at various sites in Delhi (Tables S5, S6) reveals a strong positive correlation between Bawana and JLN, Karni Singh, Najafgarh, Okhla, Patparganj, R K Puram, Rohini, and Siri Fort; Okhla-DTU; Okhla-JLN, JLN-Najafgarh, Rohini-Karni Singh, Najafgarh with Vivek Vihar and Wazirpur, Najafgarh with Okhla and Siri Fort; Okhla with Patparganj, R K Puram, Rohini, and Siri Fort; Wazirpur and Siri Fort, Wazirpur and Vivek Vihar; Rohini and Karni Singh. Similarly, a positive correlation has been observed for Ozone (Table S7) between Bawana and JLN, Karni Singh; DTU and JLN, Karni Singh; Karni Singh and Patparganj, Vivek Vihar; Najafgarh and Sri Aurobindo Marg; Patparganj and Vivek Vihar; Rohini and Vivek Vihar. Correlation analysis of ammonia (Table S8) reveals poor correlation at all sites except Bawana-Vivek Vihar and Wazirpur; and JLN-Vivek Vihar. However, if we analyze the correlation behavior among various gaseous pollutants (Table S9), NO and $NO_2$ showed a strong positive correlation with $NO_x$, indicating a common source of origin. However, these three gases revealed a slightly negative relationship with ozone gas. In contrast, lower $NO_2$ levels showed a significant negative correlation with new COVID-19 cases in Delhi by *Dutta & Jinsart (2020)*.

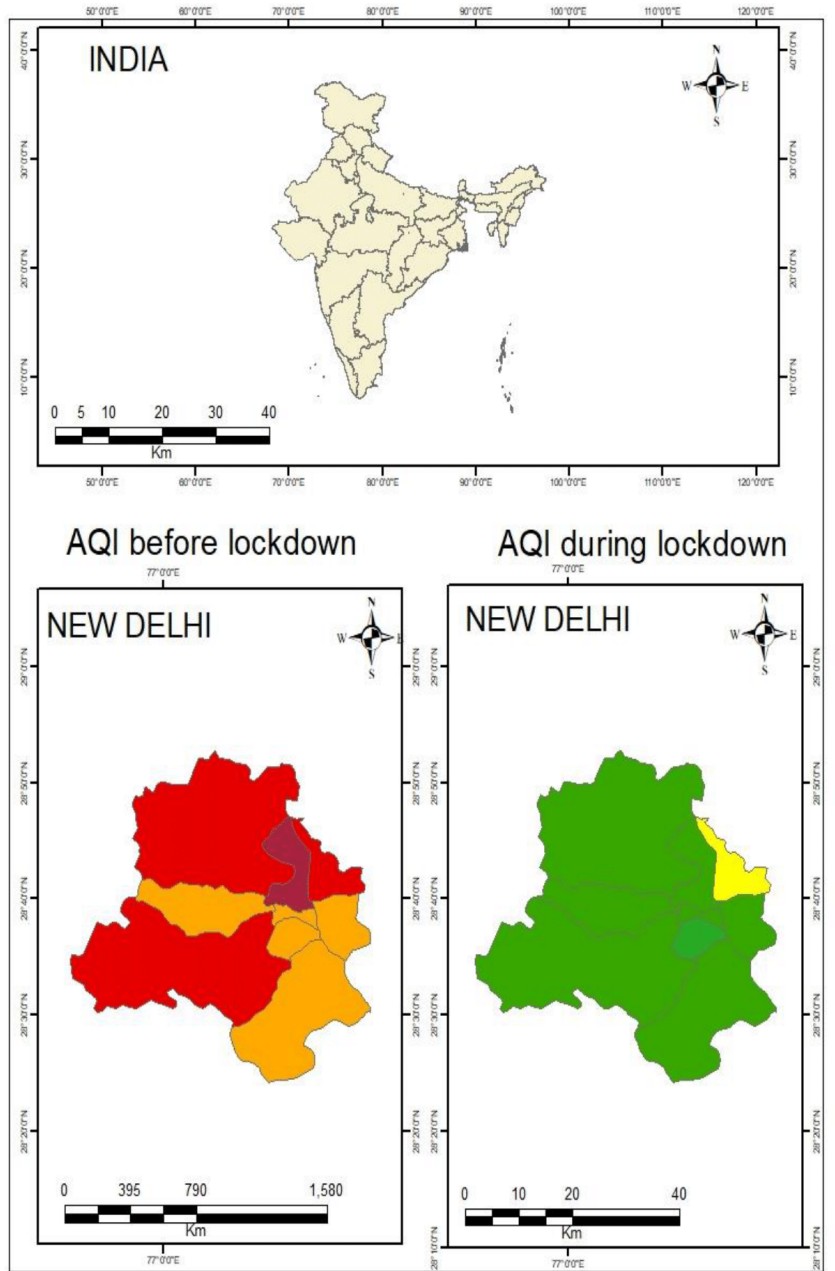

**Figure 5 Air quality before and during the pandemic in Delhi.**

## CONCLUSIONS

The results revealed a drastic improvement in air quality due to a partial lockdown measure of the National Capital of India. The main restrictions were applied from March 24 to May 30, 2020. The data for air pollutants for 21 monitoring stations was obtained from the Central Pollution Control Board (CPBC). The concentration of gaseous pollutants was significantly reduced except ozone by restricted measures in the use of private vehicles and

suspension of non-essential transportation, construction, and industrial activities during the pandemic situation. The ozone concentration was observed to be higher as compared to other pollutants during the lockdown period because of the absence of NO formation from transport emissions which could not dissociate into $O_2$ and O. The average concentrations across all stations were found to be 16.99, 27.68, 31.25, 38.55, 27.98, and 17.10 $\mu g/m^3$ for NO, $NO_2$, NOx, $O_3$, $NH_3$, and $SO_2$, respectively. The trend of gaseous pollutants zone-wise in Delhi was observed in order as NED > SWD > SD > SED > ED > ND > NoD for lockdown I and II and III whereas SED > NED > ED > SD > NoD > ND > SWD for unlockdown I and II. A dissimilar trend was observed in order as NED > ED > ND > SED > NoD > SD > SWD for unlockdown III and IV during the pandemic situation. The air quality values for pre-lockdown were found to be the "poor category for ED, NoD, SD, and SWD to "very poor" category for NoD, NED, NWD, and SED.

The health quotient (HQ) was calculated for each pollutant across the exposed groups (infant, toddler, children, and adults) for pre-lockdown, during the lockdown, and unlockdown periods in Delhi. The calculated HQ for daily and annual exposure for each pollutant across the exposed groups was less than 1 (HQ < 1) for pre-lockdown, during the lockdown, and unlockdown periods. The order of HQ for all exposed pollutants was: pre-lockdown > unlockdown > lockdown for both daily and annual exposure groups except $O_3$. The calculated HQ for daily chronic exposure for each pollutant across the infant groups was less than 1 (HQ < 1), indicating unlikely induced adverse health outcomes. However, the calculated HQ for daily chronic exposure for each pollutant across the child and adult groups was found to be more than 1 (HQ > 1), which indicated the likeliness to induce adverse health outcomes. The order of HQ for exposed pollutants was pre-lockdown > unlockdown > lockdown for both daily and annual exposure groups except $O_3$. A similar trend for annual chronic exposure was observed for all exposed groups during pre-lockdown and unlockdown periods.

The air quality levels for all zones were calculated as "Satisfactory," except for Northeast Delhi, which displayed a "moderate" category during the lockdown period. The maximum improvement in air quality was observed in Southeast Delhi (76%), and the lowest was in Northeast Delhi (46%). A correlation behavior was performed among various gaseous pollutants: NO, and $NO_2$ showed a strong positive correlation with $NO_x$, indicating a common source of origin. However, these three gases revealed a slightly negative relationship with ozone gas. On the other hand, $NO_2$ and $NO_x$ indicated a strong positive relationship with CO. The present study would serve as baseline data for future studies and also the impact of anthropogenic activities on the air quality of a region.

Limitations

- Further, increasing the scientific rigor of research in this area, some of the current manuscript's limitations require access to meteorological parameters, including rainfall, relative humidity, solar radiation, and wind speed.
- Furthermore, future studies can examine large sample sizes with different factors to draw exciting results.
- Health assessment studies for a longer time window can be calculated in future studies.

Strength

- The strength of the current manuscript was witnessed to determine changes in gaseous pollutants for pre-lockdown, during the lockdown, and unlockdown periods.
- This is the first study in Delhi, India, to estimate the health risk of human exposure to gaseous pollutants using the US Environmental Protection Agency assessment model.
- These substantial changes have been linked with human health assessment in different age groups.

In the present study, the prediction of long-term and short-term health effects in infants, children, and adults resulting from the inhalation of pollutants was possible.

## ACKNOWLEDGEMENTS

The first author thanks Ms. Saumya Kumari for insightful discussion and valuable suggestions during the preparation of the paper. The authors also appreciate Ms. Aarthi Nair and Ms. Sweety Kumari for proofreading the current manuscript.

### Funding

The authors received funding from the Researchers Supporting Project Number (RSP2022R448), King Saud University, Riyadh, Saudi Arabia. The funders had no role in study design, data collection and analysis, decision to publish, or preparation of the manuscript.

### Grant Disclosures

The following grant information was disclosed by the authors:
Researchers Supporting Project:  RSP2022R448.
King Saud University, Riyadh, Saudi Arabia.

### Competing Interests

The authors declare there are no competing interests.

### Author Contributions

- Bhupendra Singh conceived and designed the experiments, analyzed the data, prepared figures and/or tables, and approved the final draft.
- Puneeta Pandey conceived and designed the experiments, authored or reviewed drafts of the article, and approved the final draft.
- Saikh Mohammad Wabaidur performed the experiments, authored or reviewed drafts of the article, and approved the final draft.
- Ram Avtar performed the experiments, prepared figures and/or tables, and approved the final draft.
- Pramod Kumar conceived and designed the experiments, authored or reviewed drafts of the article, and approved the final draft.

- Shakilur Rahman analyzed the data, authored or reviewed drafts of the article, and approved the final draft.

## Data Availability

The raw data, such as gaseous pollutants (NO, NO2, NOx, SO2, and O3) for all the selected monitoring stations of Delhi for 24 h, is available in the Supplementary Files. They are available from the CPCB:

https://app.cpcbccr.com/ccr#/caaqm-dashboard-all/caaqm-landing/data.

## Supplemental Information

Supplemental information for this article can be found online at http://dx.doi.org/10.7717/peerj.14489#supplemental-information.

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
