# Peer review of "Substantial changes in Gaseous pollutants and health effects during COVID-19 in Delhi, India"

_PeerJ, doi:10.7717/peerj.14489_

## Round 0.1 · original submission · Major Revisions

Baed on the comments from three anonymous referees, your manuscript required major revisons on the data analysis, the reliability of inference/ conclusion, and thelanguage before it is considered to be acepted.

Reviewer 2 has suggested that you cite specific references. You are welcome to add it/them if you believe they are relevant. However, you are not required to include these citations, and if you do not include them, this will not influence my decision.

Reviewer 1 ·

Basic reporting

1. The language must be improved as the manuscript contains numerous grammatical errors.
2. Some parts are redundant. For example, the Dose-Response Assessment (2.3.2) was not performed but described.
3. “Ozone” were used in some figures, which should be revised to “O3”. Fig. 2 is very hard to read.

Experimental design

1. The method used by this work is flawed. The “Human Health Risk Assessment” part has several equations but no references. Assumptions about the population (age, weight, etc.) were not provided. It does not make sense that people of age 13-18 are missing. With a total of 9-month data, I do not think it is reasonable to calculate the annual exposure for different periods.
2. I doubt the reliability of the air pollution data used in this manuscript. How can the concentrations of NO2 (21.07 to 38.09) higher than those of NOx (15.53 to 48.77)?

Validity of the findings

This work is more like a lab report as it lacks in-depth analysis of the data. No meaningful conclusions were drawn.

Additional comments

The authors do not seem to know the basics of atmospheric chemistry. They can not understand primary and secondary air pollutants. For example, in L507, they wrote “NO formation” from transport emissions when NO is a primary pollutant.

Reviewer 2 ·

Basic reporting

The author provides important information about a new coronavirus that has harmed nearly every country's economy, ecology, and social life. Pre-, during, and after lockdowns, the short-term influence on the environment and human health must be considered, as well as the correlation between gaseous pollutants and health assessments. Manuscript is meet standareds for publication.

Experimental design

Experimental design has been done very well.

Validity of the findings

As a result, the current research shows gaseous pollutants change and their impact on air quality during the lockout. The gaseous pollutants were found to be lower in the lockdown phase than in the pre-lockdown period in the current study. This can be ascribed to the lockdown's total closure of non-traffic sources like industry and factories. The present study calculated health risk assessment and hazard identification for non-carcinogenic risks from exposure to ADD of NO2, O3, and SO2 via the inhalation route. I thoroughly loved reading the paper, The presentation is based on facts, data, and figures, I consider it complete, and the manuscript is good-looking.

Additional comments

I just have a few minor suggestions for publication of this manuscript, which I have included below. Specific comments:
1. Literature improvement: Please read the suggested studies to improve your manuscript according to these studies. Cite them in the literature and build your study objectives like these studies. These studies are very helpful and informative to enhance the quality of your work. https://doi.org/10.1080/10807039.2019.1570077
https://dx.doi.org/10.1117/12.2193542
https://doi.org/10.1016/j.apr.2019.07.004

2. Lines 99-100: You write about the number of cases and deaths due to the COVID-19 pandemic, with updated data from January 2022. However, the references presented are from the year 2022 (accessible in June 2022). I suggest checking these references and updating the current status.

3. How did you use the time series to check for changes? What adjustments were made and what period was used as the basis for the series? What are the resulting predictions? Time series results are not clearly presented.

4. Important info is missing in section 2.2. Were the data available throughout the 24 hrs? are they available for each day without interruption? What are the detection limits for the species discussed in the paper?

5. Section 2.3.4 is very poorly presented. Kindly recheck the following equation

6. line 344-348. Units are missing on line no. 417

7. Write the limitation of the current study.

Annotated reviews are not available for download in order to protect the identity of reviewers who chose to remain anonymous.

Reviewer 3 ·

Basic reporting

1. I did not find any continuity of sentences in the manuscript. Senetenses were written randomly. For example: In the result section of abstract: .... The present study calculated health risk assessment and hazard identification for non-carcinogenic. However, in infants (0-1yr), HQ for daily and annual...

2. Conclusion in Abstract starts from However, English of the manuscript needs to be improved to a greater extent.

3.To determine variable changes during the pre-, during and unlockdown periods, in addition to time series plotting technique for particulate and gaseous pollutants, it would be more pertinent to use the F-test after fitting the ARIMA to different time series data. As the title of the manuscript Substantial changes in gaseous pollutants and health effects during the COVID-19 lockdown, therefore, these changes must be based on robust statistical techniques rather than comparing the graphs only.

4. It would be better if manuscript describes the relationship between the levels of gaseous pollutants in different stages of lockdown and unlock down and health risk exposure.

5. What was the purpose to identify the correlated variables?

6. What are the limitations and strength of the manuscript?

Experimental design

no comment

Validity of the findings

no comment

Additional comments

The manuscript needs to be polished in terms of the English language, formation/continuity of sentences and statistical analysis.

---

## Round 0.2 · accepted · Accept

The manuscript has been significantly improved. The reviewer has no further specific comments. So, the manuscript can be accepted for publication.

Reviewer 1 ·

Basic reporting

No comment.

Experimental design

No comment

Validity of the findings

No comment

Additional comments

This is my second time reviewing this work which was rejected by me in the first place. It is unbelievable that the responses to the reviewers' comments are missing. I looked up my previous comments (I am so glad that I saved a copy) and read through the revised manuscript. Though I still think there is a lack of in-depth discussions, I am inclined to reserve my judgement and let the editor decided if this paper should be rejected or not.